# Endothelial Progenitor Cells as Biomarkers of Cardiovascular Pathologies: A Narrative Review

**DOI:** 10.3390/cells11101678

**Published:** 2022-05-18

**Authors:** Paul Philipp Heinisch, Corina Bello, Maximilian Y. Emmert, Thierry Carrel, Martina Dreßen, Jürgen Hörer, Bernhard Winkler, Markus M. Luedi

**Affiliations:** 1Department of Congenital and Pediatric Heart Surgery, German Heart Center Munich, School of Medicine, Technical University of Munich, 80636 Munich, Germany; hoerer@dhm.mhn.de; 2Division of Congenital and Pediatric Heart Surgery, University Hospital of Munich, Ludwig-Maximilians-Universität, 80636 Munich, Germany; 3Department of Anaesthesiology and Pain Medicine, Inselspital, Bern University Hospital, University of Bern, 3010 Bern, Switzerland; corina.bello@bluewin.ch (C.B.); markus.luedi2@insel.ch (M.M.L.); 4Department of Cardiothoracic and Vascular Surgery, German Heart Center Berlin, 13353 Berlin, Germany; max.emmert@gmx.ch; 5Institute of Regenerative Medicine (IREM), University of Zurich, 8952 Schlieren, Switzerland; 6Department of Cardiovascular Surgery, Charité Universitätsmedizin Berlin, 10117 Berlin, Germany; 7Department of Cardiac Surgery, University Hospital Zurich, 8091 Zurich, Switzerland; thierry.carrel@usz.ch; 8Department of Cardiovascular Surgery, Institute Insure, German Heart Center Munich, School of Medicine & Health, Technical University of Munich, Lazarettstrasse 36, 80636 Munich, Germany; dressen@dhm.mhn.de; 9Department of Cardiovascular Surgery, Hospital Hietzing, 1130 Vienna, Austria; bernhard.winkler@gesundheitsverbund.at

**Keywords:** endothelial cells, progenitors, cardiovascular disease, biomarker

## Abstract

Endothelial progenitor cells (EPC) may influence the integrity and stability of the vascular endothelium. The association of an altered total EPC number and function with cardiovascular diseases (CVD) and risk factors (CVF) was discussed; however, their role and applicability as biomarkers for clinical purposes have not yet been defined. Endothelial dysfunction is one of the key mechanisms in CVD. The assessment of endothelial dysfunction in vivo remains a major challenge, especially for a clinical evaluation of the need for therapeutic interventions or for primary prevention of CVD. One of the main challenges is the heterogeneity of this particular cell population. Endothelial cells (EC) can become senescent, and the majority of circulating endothelial cells (CEC) show evidence of apoptosis or necrosis. There are a few viable CECs that have properties similar to those of an endothelial progenitor cell. To use EPC levels as a biomarker for vascular function and cumulative cardiovascular risk, a correct definition of their phenotype, as well as an update on the clinical application and practicability of current isolation methods, are an urgent priority.

## 1. Introduction

Cardiovascular diseases (CVD) such as myocardial infarction (MI), cerebral and peripheral arterial disease (PAD) and arterial hypertension are the leading causes of globalmortality [1,2].

Coronary artery disease (CAD), leading to narrowing or complete blockage of arterial blood supply to the myocardium, is the most prevalent heart disease [2]. Pharmacological agents, interventional and surgical procedures, as well as diet and lifestyle-related concepts to better control established and newly discovered cardiovascular risk factors, are still not sufficient to prevent the millions of disease-related deaths worldwide every year [1,2,3,4].

Known cardiovascular risk factors (CVF) contribute to the cascade of atherogenesis especially by inducing injury and dysfunction in endothelial cells. Endothelial integrity is highly reliable due to repair and renewal by endothelial progenitor cells (EPC) derived from different sources, e.g., bone marrow (BM), circulating endothelial progenitor cells (CEPC) or adventitial residents [5]. The impaired mobilization or depletion of these cells contributes to endothelial dysfunction and CVD progression [6].

Ross’ classic paradigm already stated that endothelial cells (EC) injury is one of the most important stimuli for the development of atherosclerotic plaque [7]. In 1997, Asahara et al. reported the isolation of putative EPC from human peripheral blood, based on the cell-surface expression of CD34 and other endothelial markers and introduced the novel concept of CEPCs. These specific cells were reported to further differentiate, at least in vitro, into endothelial cells [8]. They could be identified at sites of active angiogenesis as well as in various animal models of ischemia [8]. CEPCs contribute to on-going endothelial repair through their ability to form layers of neo-endothelium at the site of injury or to serve as a cellular reservoir to replace dysfunctional endothelium [9].

Triggers for EPC recruitment in neo-angiogenesis or vascular injury include the increased availability of angiogenic growth factors or chemokines, such as the vascular endothelial growth factor (VEGF), as well as angiopoietin or stromal cell-derived factor (SDF)-1 bonding with the chemokine receptor (CXCR-4), thereby expanding its expression on EPCs [10,11,12,13]. ECs on the other hand, participate in different physiological processes, such as vasomotor tone, cellular trafficking, or innate and adaptive immunity [5]. The perioperative effects of endothelial progenitor cells, in patients with acute ischemia of the lower limbs undergoing surgical revascularization, with a plausible biological mechanism was generated for applications as a biomarker. A sound understanding and clear definition of endothelial-regenerating cells and their role in the differentiation into mature endothelium [14] is required for the establishment of an ideal test to measure EC function and enhancement of our understanding of the pathophysiology of atherosclerosis [15].

Adiponectin, adipocyte fatty acid-binding protein, heart-type fatty acid-binding protein, lipocalin-2, fibroblast growth factor 19 and 21, retinol-binding protein 4, plasminogen activator inhibitor-1, and 25-hydroxyvitamin D are just a few of the already known biomarkers linked to cardiovascular and metabolic diseases. These biomarkers may help predict CVD risk. More research is needed to assess biomarkers’ validity and their potential to improve clinical decision-making and therapy management. The endothelium has a unique position as both a sensor and participant in the atherosclerotic process [16]. Biomarkers that can be considered as indicators of biological states or conditions are progressively incorporated into cardiovascular risk assessments [17,18]. EPCs are considered the ideal target for inducing the neovascularization of ischemic tissue and can serve as a biomarkers for the surveillance of tissue damage and therapeutical outcomes [19].

In this review, we focus on current definitions of EPCs, discuss the individual relevance of circulating and adventitial resident progenitors in endothelial and vascular integrity, function, rejuvenation and restoration and address some promising therapeutic approaches and remaining questions.

## 2. Characterisation and Various Origins of Ecs/ Epcs in Humans

A consensus on the ideal marker for the identification of EPC cell types is lacking. This marker may originate from multiple precursors, such as a haemangioblast (HPC), BM progenitors, tissue-resident mesenchymal stem cells (MSC), and especially, from adipose tissue [20], impeding fast and simple isolation [5].

The three existing techniques used to isolate and cultivate human EPCs [21,22] lead to phenotypically differing cell types, and as such, potentially vague findings when analysing EPC’s role in cardiovascular repair (Figure 1) and cardiac outcome [23].

The use of density barrier centrifugation is one method that can be utilised in order to separate mononuclear cells from peripheral blood mononuclear cells (PBMNCs). In most cases, cells are sown onto plates that have been coated with fibronectin and then grown using endothelial growth factors. The remaining spindle-shaped cells not only endocytose acetylated low-density lipoprotein (LDL) but also express EC markers and possess other characteristics of ECs. This is, in addition to the fact that EC indicators are present in these cells [24], acquired via antigen transfer from platelets that contaminate isolates of PBMNCs [25]. Platelets on mononuclear cell cultures degrade into micro-particles (vesicles that retain specific antigens from the cell of origin) within 7 days, and CD31 expression (along with platelet-specific markers) was present at that time on EPCs, now called putative EPC’-aggregates‘, whereas EPCs were CD31-negative on day 1 [25]. Attempts to further purify the cell cultures led to the establishment of new protocols.

Using markers such as CD133, combined with CD34 and VEGFR2, ensured that only progenitor cells with vasculogenic properties were identified [26,27]. Cell labelling with antigen-specific antibodies and fluorescence-activated cell sorting (FACS) to select EPCs was applied [8], while still culturing the cells on fibronectin [25]. The CD34+ cells were surrounded by spindle-shaped cells expressing increased EC markers. Through the use of this marker combination, this cell type was successfully isolated from adult peripheral blood, umbilical cord blood, and foetal liver [28]. Recent research shows that human CD34+/CD133+/VEGFR2-positive cells are separate primitive haematopoietic progenitors lacking vessel formation ability and expressing CD45, a haemangioblast marker [29,30]

Thereafter, in vitro colony-forming cell assays allowed for the isolation of two cell types: colony-forming unit (CFU)-Hill cells and endothelial colony-forming cells (ECFC). In the ‘Hill assay’ and the ECFC, or ‘Ingram protocol’, monocytic cells isolated from blood samples were cultured for two days on fibronectin-coated dishes and then replated for further cultivation [6,27]. CFU-Hill cells are phagocytic and express EC-like markers (CD14, CD45, and CD115) but lack proliferative and vasculogenic activity. While lacking CD14, CD45, and CD115, ECFCs express EC markers and have the ability to form capillary-like structures in vitro and vessels in vivo [21]. ECFCs have been shown to reside in the arterial wall suggesting that this may be the main origin of these cells [26].

In summary, EPCs seem to represent two distinct populations with overlapping antigen expressions (e.g., CD34/VEGFR2): hematopoietic-derived spindle-shaped cells from isolation method I/II, also referred to as circulating angiogenic cells or early EPCs (CFU-Hill colonies), and ECFCs, or late EPCs [31]. Late EPCs have a vasculogenic ability in vitro and are well-integrated into membranes, whereas early EPCs act via a paracrine mechanisms [32] and might even protect late EPCs from oxidative stress [33].

## 3. Influence on Vascular Pathologies and Role as a Biomarker

At present, there is a lack of information or knowledge regarding cell phenotypes in different diseases as the cells originate from different vascular beds and sources [34]. Half of the CECs from healthy controls express CD36, a marker for cells of microvascular origin, whereas in sickle cell anaemia, this percentage increases to 80% [35]. Contrastingly, no CD36 could be stained in CECs from patients with acute coronary syndrome, consistent with the macro-vascular origin of these cells [36].

Investigating the role of CECs in endothelial injury with regard to plasma markers of endothelial injury (vWf, tissue plasminogen activator, soluble E-selectin) led to a correlation between CECs and vWf in heart failure [37].

Almost all types of CVD were associated with hypertension, diabetes, smoking and high cholesterol. These CVFs can contribute to endothelial dysfunction [38,39,40,41,42,43,44,45,46,47,48]. High homocysteine and ADMA values also showed a negative effect on EPC count [49,50]. On the other hand, high HDL cholesterol and TG levels correlated with CFU but not with CD34/133+ cell count [51]. (Table 1) Statin [52] and Angiotensin receptor II inhibitors [53], as well as oestrogen levels (high oestrogen levels in women were associated with an increased EPC count [54] in animal carotid injury oestrogen-enhanced EPC function [55]), glitazones [56], erythropoietin [53,57,58,59], and PDE5 inhibitors [60] all showed beneficial effects. EPC count was also dependent on SDF-1 [61,62], VEGF [10,11], NO [63], GCS-F and GM-CSF [64,65] levels.

Physical activity at a moderate level was identified to be potentially beneficial for preventing CVD. The increase in EPC count was found to be mediated by eNOS and VEGF, and apoptosis was reduced in the cells [66]. Physical activity lead to a higher amount of circulating CD34-positive EPCs in CVD patients [67]. Furthermore, this study identified an association of CD34-positive cell count with lower all-cause and cardiovascular mortality [67]. Vascular damage progression correlated with EPC count [68] just as a decreased amount of CD34-positive but increased amount of CD34^+^CD133^+^CD309^+^ and CD34^+^CD133^+^ cells suggested the progression of cerebral small vessel disease [69]. EPC count could, therefore, serve as a biomarker for CVD course. A correlation with CAD progression was also found for osteocalcin, a regulator of early EPC. A higher number of CVRFs was associated with a decreased total osteocalcin count. Osteocalcin positivity in EPCs was related to LDL, total cholesterol and TGs in both early and, significantly, in late CAD [70]. EPC count could also be used as a marker in treatment monitoring, such as in chronic total coronary artery occlusion since an association with Rentrop grade at baseline and 1 year post operation was discovered [71].

Some data still suggest that those “monitoring effects” are mostly seen in the young population. Aging by itself is another depriving factor of EPC [6,63]. Age directly limits EPC mobilisation but also via VEGF depletion and physiologically lowered NO levels, which contribute to the bad survival and proliferation of EPCs [63,72]. Moreover, several mechanisms, including the co-existence of CVF, lead to the impaired maintenance of endothelial integrity [73]. In elderly CAD patients with stable disease, EPC count was significantly reduced compared to younger patients [74]. The mobilisation of EPCs was also lower after a coronary bypass grafting in advanced age [75,76]. The severity of stable CAD shows an inverse correlation with the total/early EPC number [62,77]. Additionally, chronic vascular disease appears to have opposite effects on early and late EPC numbers and does not influence their functional capacity [62,77].

Using EPC as a therapeutic target for CVD may therefore underlay an age-related effect. Early EPC implantation increased neovascularisation in young mice, but not in older mice with elevated cholesterol or other CVD risk factors [78].

## 4. The Role of Epc in Congenital Heart Disease Heart Failure

Mechanisms of heart failure in adult heart disease recently received a lot of research attention, but its pathogenic and prognostic significance in single-ventricle physiology is still unknown [79,80,81]. Congenital cardiac malformations with a single ventricle have a high risk of mortality in the first year of life in such patients and frequently result in late complications developing during this stage of palliative repair [82]. Even though single-ventricle reconstruction trials have sought to identify predictors of poor outcomes at three years in patients with single-ventricle physiology based on the types of initial shunt (Norwood procedure with ventriculo-pulmonary Sano shunt or with modified subclavio-pulmonary Blalock shunt) and the timing of stage 2 palliation, a 12-year longitudinal cohort study in patients with Fontan (stage 3 procedure) circulation found that the risk of death or cardiac transplantation was closely associated with poorer ventricular function [83,84]. No definitive therapy was shown to improve heart function with a chronic volume or pressure overload, which may worsen prognosis for single-ventricle patients [85].

However, early phase 1/2 clinical trials utilizing the intracoronary delivery of derived progenitor cells demonstrated dependable and safe outcomes in patients with single ventricle physiology. Except for all-cause mortality after staged procedures, derived cardiac progenitors cells administration improved ventricular function and was linked with fewer late problems in patients with single ventricles. Patients treated with cardiac progenitors cells and those who suffered from heart failure with reduced EF but not heart failure with intact EF may experience a substantial improvement in clinical outcomes [86,87].

## 5. Targeting of Treatment for CVD

Nonetheless, increasing the EPC count is a promising strategy. Early information [8] on EPC’s contribution to neo-angiogenesis could be secured in mice models of ischemia, where CD34- and stem cell antigen–1 (Sca-1)-enriched populations of mononuclear cells promoted new blood vessels and enhanced the perfusion and initiation of recovery in ischaemic tissue [64]. Kong et al. (2004) observed endothelial repair and neointima development after the cytokine-induced mobilization of circulating progenitors [88]. After EC injury, spleen-derived mononuclear cells and cultivated early EPCs, but not BM-derived EPCs, increased re-endothelialisation and decreased neointima formation [89]. In several models of vascular graft atherosclerosis, recipient-circulating progenitor cells were required to create an endothelial monolayer. An allograft of mouse Balb/c aorta into the carotid artery of chimera mice with BM from Tie2-LacZ mice showed activity 4 weeks following surgery. The quantification of the obtained data indicated that more than 70% of the regenerated endothelium was derived from non-BM tissues [90,91,92]. In humans, the implantation of bone marrow mononuclear cells (BMMCs) was found to be an effective treatment for PAD in an attempt to alleviate limb ischemia through the use of stem cells. Angiogenesis occurred as a result of the stimulation of EPC and the release of VEGF and cytokines [93]. The promotion of EPC and the release of VEGF and cytokines alleviated the symptoms and collateral circulation [93].

Yet, there are dangers associated with the direct implantation of EPCs mainly triggered by the high number of cells with inflammatory potential that could be coinjected [94]. Despite different routes of administration that are considered safe, such as intracardiac, intrahepatic or intramuscular [95], indirectly targeting EPC count and function might be a promising treatment option, lowering the associated risks of cell transplantation. CD-34-coated stents were applied and compared to paclitaxel, with good outcomes and significantly less post-discharge thrombotic events after 12 months follow-up [96]. In diabetic cardiomyopathy, treatment with BMP7, an anti-inflammatory protein, led to an increase in EPC markers and neovascularisation, ultimately improving cardiac remodelling [97]. Blocking hormonal pathways that are important in CVD development, such as gonadotropin, an enhancer of ECFC angiogenesis, and thrombin might also be beneficial [98]. Using EPC-derived vesicles [99] or exosomes [100] further helps to mitigate the problematic isolation and culturing process of EPCs, thereby providing more efficient therapeutic options.

Additionally, the transfer of plasmid DNA for VEGF to young and senescent EPCs via ultrasonic microbubble transfection could enhance the angiogenic effects of both older and young cells, thereby leading to better outcomes and solving two problems at once [101]

Other new therapeutic options with EPC as a target include the treatment of ischemic stroke in patients with pre-existent CVD [102].

## 6. Ongoing Challenges

There is a high risk of postoperative morbidity for patients following major surgery. Endothelial dysfunction in the perioperative period may increase the risk of surgical complications through altered vascular homeostasis and, consequently, decrease tissue perfusion. Targeting the endothelium and optimising natural physiological function in the perioperative phase is becoming a more popular as a way to improve postoperative outcomes. With an estimated surface area of more than 1000 m^2^, the human vascular system constantly maintains a balance between coagulation and bleeding, inflammation and immuno-protection rendering difficult the identification of one single area of activation or damage contributing to the release of a specific set of ECs [103]. MSCs and EPCs differ in their immunomodulatory and immunosuppressive effects. MSCs and EPCs reduce T-cell proliferation, activation and cytokine production and MSCs, in comparison to EPCs, may induce regulatory T-cells [104]. EPCs with haemangioblast properties (CD34+ and CD45+) or “early EPCs” do not differentiate into ECs supporting the fact that “true EPCs” reside in the vasculature [105]. Their role in vascular repair of ischemic tissues might rather be via paracrine mechanisms, such as the secretion of angiogenic cytokines (e.g., VEGF) [5,32].

Differentiating EPCs from CECs remains difficult [34]. Endothelial markers, such as CD-146 and UEA-1 are present on EPCs and severely ‘morphology-damaged’ cells obtained by CD-146-driven immune–magnetic isolation alike. The inability to grow these cells in culture is still regarded as an implication that these cells are not EPCs [106]. The activation of ECs occurs by pro-inflammatory cytokines, growth factors, lipoproteins, or even oxidative stress. Therefore, detected detached cells might be apoptotic or necrotic, distorting the set of detected circulating cells [107] and further confounding isolates of EPCs.

While the identification of resident-tissue EPCs is advancing somewhat, the mechanism of progenitor cell release and the distinct role of different EPC types are still not understood. Differentiating between the role of early and late EPCs in CVD is important as, in CAD patients, a decreased amount of early EPCs was found [108], accompanied with higher density of late EPCs [109].

Most studies to date describe factors which promote/inhibit either total (CD34+ or CD34/VEGFR2) or early EPC (CD34+/VEGFR2 +/CD133+ or VEGFR2+/CD133+) mobilisation and homing from the BM under physiological and pathological drug therapy conditions [110]. Changes in EPC count could be registered in patients displaying an onset of acute ischemia following MI [111,112,113], unstable angina, coronary artery bypass grafting [75,76] or stent implantation [114,115].

Within the identification of all of these influencing factors, a single clear mechanistic explanation to understand EPC function, mobilisation and endothelial integrity is lacking. Improvements in risk classification and the ability to regulate thrombotic and inflammatory cascades should enhance perioperative outcomes. The underlying subclinical microvascular endothelial dysfunction seems to have a greater impact on perioperative morbidity, contributing more to complications, such as impaired wound healing and end organ dysfunction, than the less common but more devastating macrovascular endothelial dysfunction.

Ongoing studies address sheer stress as another key factor since low or disturbed shear stress, which occurs in vessel branch points, the outer wall of bifurcations, and the inner wall of curvatures are considered pro-atherogenic [116]. Cell turnover rates in the arteries are very low, but in some areas this corresponds to increased permeability to plasma proteins [117], in atherosclerotic-lesion-prone site [118], and in areas of low shear stress, a high endothelial death rate and overall high turnover rate is needed to maintain vessel homeostasis [116]. Therefore, recent theories addressed the role of different flow patterns on EPC function [119]. A disturbed flow was associated with a higher mitotic and apoptotic activity of ECs and lessened eNOS expression [119]. Long-time outcomes of stenting were determined by endothelial cell turnover or number [120]. Histone deacetylases (HDAC) were identified as key players behind this finding [120]. A disturbed flow induced the transient stabilization of the HDAC3 protein in ECs by stimulating VEGFR2 [121]. HDAC3 belongs to the class I HDACs that enhances the removal of acetyl groups from histone and non-histone proteins [120,122]. HDAC3 is critical for EC survival and acts as a flow-pattern-dependent pro-survival molecule [123]. However, HDAC7 is modulated by VEGF and turbulent flow and is involved in endothelial homeostasis and differentiation, as well as vascular SMC proliferation [124]. SMC play the main role in the pathogenesis of vascular-disease-mediated restenosis [124]. Aside from being a potential drug target, HDACs could help explain post-interventional outcome differences. Trichostatin, for example, inhibits HDAC, which may be useful in future CVD treatments [125]. Such important players in CVD pathophysiology should be studied more extensively within the context of EC research in order to better understand the mechanisms involved and ultimately find new targets of treatment and biomarkers to monitor disease and measure therapeutic outcomes.

## 7. Conclusions

EC architecture and mechanisms of endothelial homeostasis are potential sensors and target therapeutic interventions in the atherosclerotic process. While current studies assessing EC subunits, phenotype and function enabled a general understanding of the pathophysiology of atherosclerosis, an ideal test of endothelial function for use in a clinical setting has yet to be established. A secure method for counting cells and differentiating cell kinds and compositions is urgently required, as is a deeper understanding of other modulators such as flow pattern, growth and transcription factors, and gene expression. CECs may serve as sensitive indicators of pre-existing damage and disease development, whereas EPCs may act as a biomarker of repair and a promising therapeutic target. Regenerative medicine has the potential to expand the therapeutic window for a variety of diseases, where surgical and medicamentous alternatives have run out. In CVD, safe cell replacement with new ones could usher in a new era of therapy.

## Figures and Tables

**Figure 1 cells-11-01678-f001:**
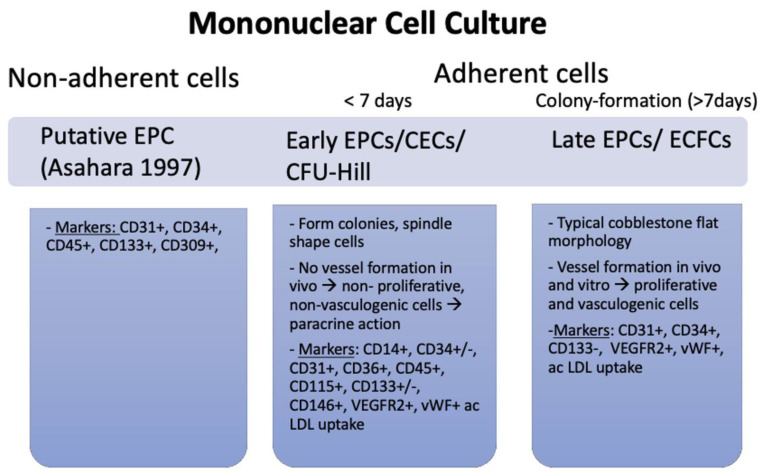
Three different types of EPCs, their function and associated markers. In Asahara et al.’s method, non-adherent cells were cultivated, whereas in the protocols by CFU-Hill and Ingram, only adherent cells were used. The three cell cultures each have a distinct morphology, specific functions in vivo and display discriminating markers [8].

**Table 1 cells-11-01678-t001:** Correlations of EPC count, EPC function and EPC apoptosis with cardiovascular risk and protective factors, pathophysiologic state, physiologic mediators and common drugs in cardiovascular disease.

	EPC Count	EPC Function	EPC Apoptosis
Decreased	CVF: hypertension, diabetes, smoking, high cholesterol, high ADMA values, high homocysteine; Vascular damage progression; Severity of CAD aging; Chronic vascular disease.	Aging (mobilisation, function, integrity); Coronary artery bypass grafting.	Physical activity
Enhanced	Statin, ARBs, oestrogen; Physical activity (via eNOS, VEGF).	Oestrogen, glitazone, erythropoietin, PDE5 inhibitors, SDF-1 (mobilisation), VEGF, GCS-F, GM-CSF (proliferation).	Aging

## Data Availability

Not available.

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
