# Peer review of "Endothelial Progenitor Cells as Biomarkers of Cardiovascular Pathologies: A Narrative Review"

_cells, 2022, doi:10.3390/cells11101678_

Round 1

Reviewer 1 Report

Comments to the Authors

Review:

“Endothelial Progenitor Cells as biomarkers of cardiovascular pathologies. A narrative review”

I appreciate the opportunity to review the article by Heinisch et al. entitled “Endothelial Progenitor Cells as biomarkers of cardiovascular pathologies. A narrative review”

First of all, I want to thank the authors for submitting their work to “Cells” and want to congratulate them for a well written manuscript.

Please find below some comments/thoughts which might help to further improve this manuscript:

  • General comment
    • The introduction is well written and the aims of the review are clearly stated. Since this manuscript was submitted for the Special Issue: “Biomarkers and Cellular Biology in Perioperative Medicine”, a few words associated to the impact of Endothelial (Progenitor) Cells in the perioperative setting could further improve the Introduction section, especially for clinicians involved in the perioperative management of patients.
    • In my humble opinion, the same applies for the other sections, e.g. “ongoing challenges”. Where do the authors see the biggest potential of EPCs? Maybe this could also be focused more on the perioperative setting to address the special issue.
  • Minor Comments
    • Line 77 and 82: Please write out abbreviations when first used, e.g. EC
    • Line 81: Reduce space between “least” and “in-vitro”
    • Line 304: Please remove “survival after or?”

Author Response

Thank you very much for your comments on our manuscript. Thank you very much for giving us the opportunity to revise our manuscript. We have responded to all comments as listed below. Changes in the text are highlighted in red.

We added a section to the "introduction" and "ongoing challenges to highlight the potential use of endothelial progenitors cells in the perioperative setting.

Introduction Lines 91-94: "The perioperative effects of endothelial progenitor cells in patients with acute ischemia of the lower limbs undergoing surgical revascularization, with a plausible biological mechanism generated for application as a biomarker."

  Discussion Lines 268-272: "There is a high risk of postoperative morbidity for patients following major surgery. Endothelial dysfunction in the perioperative period may increase the risk of surgical complications through altered vascular homeostasis and consequently decreased tissue perfusion. Targeting the endothelium and optimising natural physiological function in the perioperative phase is becoming more popular as a way to improve postoperative outcomes."   Discussion Lines 303-308: "Improvements in risk classification and the ability to regulate thrombotic and inflammatory cascades should enhance perioperative outcomes. The underlying subclinical microvascular endothelial dysfunction seems to have a greater impact on perioperative morbidity, contributing to complications like impaired wound healing and end organ dysfunction, than the less common, but more devastating, macrovascular endothelial dysfunction."     Minor Comments:  
  • Line 77 and 82: Please write out abbreviations when first used, e.g. EC
    • Changed accordingly by the authors
  • Line 81: Reduce space between “least” and “in-vitro”
    • Changed accordingly by the authors
  • Line 304: Please remove “survival after or?” 
    • Changed accordingly by the authors
    •  

Reviewer 2 Report

In this review by Paul Philipp Heinisch et al., the authors focus on Endothelial Progenitor Cells as biomarkers of cardiovascular pathologies. Overall, the review is well written and cover the literature of the last years.

  1. Biomarkers of endothelial dysfunction can be classified in traditional ones and novel circulating biomarkers such as chemokines, extracellular vesicles. Moreover the innovative introduction of omics technologies has become a potential tool for novel biomarkers such as miRNA, metabolites. I would suggest adding in the introduction a brief description of other relevant biomarkers of cardiovascular pathologies.
  2. Page 2 line 91-95 please re-phrase this sentence. The part “monitoring or control of treatment” is not clear to me.
  3. Page 4 line 153-154 what does it mean “in overcoming recent problems in tissue engineering”? This part is not clear. How EPC can be useful in tissue engineering? Please better explain this concept.
  4. Page 5 line 190-193 please better write this sentence, it is not clear and not written in proper language.
  5. Page 7 line 304 there is a part not deleted.

Author Response

Thank you very much for giving us the opportunity to revise our manuscript. We have responded to all comments as listed below. Changes in the text are highlighted in red.

1. Biomarkers of endothelial dysfunction can be classified in traditional ones and novel circulating biomarkers such as chemokines, extracellular vesicles. Moreover the innovative introduction of omics technologies has become a potential tool for novel biomarkers such as miRNA, metabolites. I would suggest adding in the introduction a brief description of other relevant biomarkers of cardiovascular pathologies.

Response: We added a section to the introduction with a focus on different biomarker.

Change: "Adiponectin, adipocyte fatty acid-binding protein, heart-type fatty acid-binding protein, lipocalin-2, fibroblast growth factor 19 and 21, retinol-binding protein 4, plasminogen activator inhibitor-1, and 25-hydroxyvitamin D are just a few of the already known biomarkers linked to cardiovascular and metabolic diseases. These biomarkers may help predict CVD risk. More research is needed to assess biomarkers' validity and their potential to improve clinical decision-making and therapy management."

2. Page 2 line 91-95 please re-phrase this sentence. The part “monitoring or control of treatment” is not clear to me.

Response: Thank you of your comment. The sentence was changed

Change Introduction: "A sound understanding and clear definition of endothelial-regenerating cells and their role in differentiation into mature endothelium14is required for the establishment of an ideal test to measure EC function and enhancement of our understanding of the pathophysiology of atherosclerosis.15"

3. Page 4 line 153-154 what does it mean “in overcoming recent problems in tissue engineering”? This part is not clear. How EPC can be useful in tissue engineering? Please better explain this concept.

Response: Thank you of your comment. The sentence was misleading and deleted from the section.

4. Page 5 line 190-193 please better write this sentence, it is not clear and not written in proper language.

Response: Thank you very much for the comment.

Change: "EPC count could also be used as a marker in treatment monitoring, such as in chronic total coronary artery occlusion, as an association with Rentrop grade at baseline and 1 year postoperatively was discovered."

5. Page 7 line 304 there is a part not deleted.

Response: Thank you very much for the comment. The part was deleted accordingly.

Change: "Long-time outcome of stenting was determined by endothelial cell turn over or number."

Reviewer 3 Report

The manuscript “Endothelial Progenitor Cells as biomarkers of cardiovascular pathologies. A narrative review” refers to an important topic. Moreover, data regarding EPCs are still relatively scarce.

Remarks:

1) Paragraph: 118-129 in the chapter: Characterisation and various origins of ECs/ EPCs in the human). It is unclear why the authors describe this protocol and quote “Zhang SJ, Zhang H, Hou M, et al. Stem Cells Dev 2007;16(4):683-90. DOI: 10.1089/scd.2006.0062” who state: “In conclusion, adult bone marrow-derived LDL uptake-positive cells that have been reported so far actually are monocytes/macrophages that can express some endothelial markers but are not "true endothelial progenitor cells" (EPCs). MSCs, which are the only cell type that shows strong proliferation during long-term adherent culture for bone marrow cells, do not differentiate toward the endothelial lineage when grown under endothelial promoting conditions.”

Please either explain this or rewrite this paragraph.

2) Table 1. Please add references.

3) Lines 332-227 – please fill those lines.

Author Response

Thank you very much for giving us the opportunity to revise our manuscript. We have responded to all comments as listed below. Changes in the text are highlighted in red.

1. Paragraph: 118-129 in the chapter: Characterisation and various origins of ECs/ EPCs in the human). It is unclear why the authors describe this protocol and quote “Zhang SJ, Zhang H, Hou M, et al. Stem Cells Dev 2007;16(4):683-90. DOI: 10.1089/scd.2006.0062” who state: “In conclusion, adult bone marrow-derived LDL uptake-positive cells that have been reported so far actually are monocytes/macrophages that can express some endothelial markers but are not "true endothelial progenitor cells" (EPCs). MSCs, which are the only cell type that shows strong proliferation during long-term adherent culture for bone marrow cells, do not differentiate toward the endothelial lineage when grown under endothelial promoting conditions.”

Response: We greatly appreciate your feedback on our manuscript. We added the Zhang et al. study due to the investigation into the various EPC types. Two EPCs were detected in the peripheral blood of humans. There have been previous reports on these two types of EPC, but this is the first time that they have been compared and displayed together. Comparing the morphology, proliferation rate, and survival of two EPC types. They had distinct profiles of gene expression, resulting in distinct in vitro functions. Early and late EPC contributed to neovasculogenesis in vivo by secreting angiogenic cytokines and providing endothelial cells, despite their differences in gene expression and function in vitro.

Change: "

The use of density barrier centrifugation is one method that can be utilised in order to separate mononuclear cells from peripheral blood mononuclear cells (PBMNCs). In most cases, the cells are sown onto plates that have been coated with fibronectin and then grown using endothelial growth factors. The remaining spindle-shaped cells not only endocytose acetylated low density lipoprotein (LDL) but also express EC markers and possess other characteristics of ECs. This is in addition to the fact that EC indicators are present in these cells. "   2. Table 1. Please add references.   Response: Thank you for the comment. We added the reference to the section.   3. Lines 332-227 – please fill those lines.   Response: Thank you for the comment. We added the information  to the sections.